# Classical and Quantum Signatures of Quantum Phase Transitions in a (Pseudo) Relativistic Many-Body System

**Maximilian Nitsch \*, Benjamin Geiger \*, Klaus Richter \* and Juan-Diego Urbina \***

Institut für Theroretische Physik, Universität Regensburg, 93040 Regensburg, Germany

\*   Correspondence: maximilian.nitsch@physik.uni-regensburg.de (M.N.);
    benjamin.geiger@physik.uni-regensburg.de (B.G.); klaus.richter@physik.uni-regensburg.de (K.R.);
    juan-diego.urbina@physik.uni-regensburg.de (J.-D.U.)

**Abstract:** We identify a (pseudo) relativistic spin-dependent analogue of the celebrated quantum phase transition driven by the formation of a bright soliton in attractive one-dimensional bosonic gases. In this new scenario, due to the simultaneous existence of the linear dispersion and the bosonic nature of the system, special care must be taken with the choice of energy region where the transition takes place. Still, due to a crucial adiabatic separation of scales, and identified through extensive numerical diagonalization, a suitable effective model describing the transition is found. The corresponding mean-field analysis based on this effective model provides accurate predictions for the location of the quantum phase transition when compared against extensive numerical simulations. Furthermore, we numerically investigate the dynamical exponents characterizing the approach from its finite-size precursors to the sharp quantum phase transition in the thermodynamic limit.

**Keywords:** phase transitions; semiclassical approximation; Dirac bosons; mean field analysis; adiabatic separation

## 1. Introduction

The methods and ideas of quantum chaos [1,2] provided deep insights into the way classical information conspires with $\hbar$ in a subtle manner. To a large extent, this can be understood within a semiclassical theory, explaining genuine quantum behaviour, such as entanglement and coherence. In this field, Shmuel Fishman made paramount contributions ranging from the celebrated explanation of dynamical localization as a type of Anderson transition in kicked systems [3] to the resummation of periodic orbit expansions to construct semiclassical approximations for individual eigenstates in chaotic systems [4]. The present contribution aims to express our admiration for his scientific work.

During the last decade, the field of quantum chaos experienced an influx of new ideas coming from its application to the realm of interacting many-body systems. The newly emerging field of many-body quantum chaos is based on exciting developments in our understanding of fundamental problems, such as the equilibration of closed systems [5–9] and the scrambling of quantum information due to classical chaos [10–13].

It is, therefore, not a surprise that semiclassical methods, both at the heuristic level of quantum-classical correspondence [14–16] and the level of asymptotic analysis of path integrals describing coherent quantum effects [17–21], were lifted from their original particle-like form into the realm of quantum fields. Among the plethora of phenomena characteristic of the rich physics of interacting many-body systems, critical phenomena have always had a special place. In this new disguise, many-body semiclassical methods are a suitable tool to understand even the most delicate quantum effects related to the emergence of criticality.

A natural arena for testing this idea is the attractive Lieb-Liniger model [22] describing one-dimensional bosons attractively interacting through short-range forces and, in particular, its low-energy effective description that was experimentally realized [23,24]. The reason for this is that this system displays a quantum phase transition [25–28] and admits a proper semiclassical derivation of a well-defined and controlled classical limit in the form of mean-field equations, thus allowing for direct application of semiclassical techniques [21]. The semiclassical study of this system in [21] revealed the key role played by locally unstable mean-field dynamics in the corresponding dynamical and spectral quantum mechanical features.

The extension of many-body semiclassics beyond the realm of bosonic systems is still in its infancy, but a step in this direction is to first consider how the well-established picture of [21] gets modified by two new ingredients: a relativistic dispersion and the presence of spin-like degrees of freedom. Since the very possibility of having locally unstable dynamics (as opposed to global chaos) of the attractive Lieb-Liniger model is due to the integrability of the effective Hamiltonian describing its low energy regime, a natural question concerns possible non-integrable behaviour of such models and its consequences for the existence and characteristics of the quantum phase transition. In this paper, we answer some of these questions.

The paper is organized as follows. After we introduce the model and describe its general physical properties in Section 2, we present the motivation for the transformation into a special Fock basis in Section 3 and how this optimal transformation adiabatically fragments the Hamiltonian in Section 4. After that, in Section 5, the conversion of the channel containing the ground state into its classical form is examined. The most important results presented in the Section 6 are the exact calculation of the critical interaction strength and the analysis of discontinuities in the functional dependence of the energy on the interaction. Finally, the asymptotic convergence of the first excited energy level towards the ground state level leading to a degenerate ground state in the mean field is quantified in Section 7.

## 2. The Hamiltonian and Its Symmetries

The Hamiltonian of the (modified) Lieb-Liniger model with linear dispersion and contact potential is defined as

$$\hat{H} = -i\hbar \sum_{\beta=1}^{N} \hat{\partial}_\beta \otimes \hat{\sigma}_z^{(\beta)} - \frac{R\alpha}{4} \sum_{\beta,\gamma=1}^{N} \delta(\hat{x}_\beta - \hat{x}_\gamma)(\hat{\sigma}_x^{(\beta)} + \hat{\sigma}_x^{(\gamma)}), \tag{1}$$

describing bosons on a ring with radius $R$ with a contact interaction that can be interpreted as a mass term: The moment two bosons are at the same point they obtain a mass through the contact potential, whereas they are massless otherwise. In the following we assume attractive interactions, e.g., $\alpha > 0$, and we will choose natural variables $\hbar = 1$, $L = 2\pi R = 2\pi$ such that the unit of energy is $[E] = \frac{\hbar}{R}$ [28].

As appealing as it is, it is important to note that the system above appears ill-defined, as its Hamiltonian (1) is not bounded from below. Unlike in fermionic systems, in this bosonic system this issue cannot be resolved by the introduction of a Fermi sea. One way out of the problem is to interpret (1) as emerging from a local approximation of a one-dimensional condensed matter or cold atom system with two crossing bands that is perturbed by an interband interaction. This naturally introduces a regularization of the noninteracting model with a single-particle momentum cutoff defining the region where the linearization is justified. In this approach, the linear dispersion is a property of excited states and has an effect on dynamical properties of states with a certain momentum. An example of such (local) Dirac bosons in two dimensions was found in the collective plasmon dispersion relation in honeycomb-lattices of metallic nanoparticles [29]. In such local approximation, one has to make sure that any prediction of the model has to be independent of the cutoff, which might be realized in a quench scenario, starting with a narrow momentum distribution.

However, we take a different perspective here that takes a truncated model as it is, i.e., we truncate to the three lowest single-particle momentum modes (for each quasi-spin, see below) and then assume that the ground state of this model represents a physical ground state. One possible realization of such a system is obtained by mapping the truncated model to a spin-one bose gas on two quantum

dots (or two sites with suppressed hopping), where the physical spin takes the role of the momentum $k = -1, 0, 1$ and the pseudo-spin 1/2 labels the two sites that have opposite external magnetic fields applied to them, introducing linear Zeeman splitting and thus the "three-mode linear dispersion". The interaction processes are then taking, e.g., two particles of opposite spin on the same site and distribute them into the spin-zero modes of the two sites. There are different processes, of course, but the overall interaction effect is a spin-mediated hopping of a single particle with the total spin (of the participating particles) being preserved. The noninteracting case would decouple the two sites.

To implement the truncation within a Fock space approach, we choose the eigenbasis of the non-interacting ($\alpha = 0$) Hamiltonian as the single-particle basis

$$|k, \sigma\rangle = |k\rangle \otimes |\sigma\rangle\,, \tag{2}$$

where as orthonormal eigenbasis for the momentum operator we use plane waves

$$\langle x|k\rangle = \frac{1}{\sqrt{2\pi}} e^{ikx}, \text{ with } k \in \mathbb{Z} \tag{3}$$

as the most obvious choice. For the quasi-spin an orthonormal eigenbasis is used consisting only of "up" and "down"

$$\sigma \in \{+1, -1\}, \ |\sigma\rangle \in \left\{ \begin{pmatrix} 1 \\ 0 \end{pmatrix}, \begin{pmatrix} 0 \\ 1 \end{pmatrix} \right\} \tag{4}$$

generated by the third Pauli-matrix

$$\hat{\sigma}_z |\sigma\rangle = \sigma |\sigma\rangle\,. \tag{5}$$

From these definitions the Fock space is characterized through the occupation numbers $n_{k,\sigma}$ of the several states $|k, \sigma\rangle$ with creation and annihilation operators satisfying canonical commutation relations

$$[\hat{a}_{k,\sigma}, \hat{a}_{l,\tau}^\dagger] = \delta_{k,l}\delta_{\sigma,\tau}, \qquad [\hat{a}_{k,\sigma}, \hat{a}_{l,\tau}] = 0, \qquad [\hat{a}_{k,\sigma}^\dagger, \hat{a}_{l,\tau}^\dagger] = 0, \tag{6}$$

where each pair of creation/annihilation operators defines an occupation number operator

$$l\hat{n}_{k,\sigma} = \hat{a}_{k,\sigma}^\dagger \hat{a}_{k,\sigma} \tag{7}$$

for the corresponding mode. With the help of these bosonic operators this leads, after truncation of the momenta from $\mathbb{Z}$ to $\{-1, 0, 1\}$, to the more convenient form

$$\hat{H} = \sum_{\substack{k \in \{-1,0,1\} \\ \sigma \in \{-,+\}}} \sigma k \cdot \hat{a}_{k,\sigma}^\dagger \hat{a}_{k,\sigma} - \frac{\alpha}{2} \sum_{\substack{k,l,m,n \in \{-1,0,1\} \\ \sigma,\tau \in \{-,+\}}} \hat{a}_{k,\sigma}^\dagger \hat{a}_{l,\tau}^\dagger \hat{a}_{m,-\sigma} \hat{a}_{n,\tau} \cdot \delta_{k+l,m+n}, \tag{8}$$

with the relevant Fock states labeled by six occupation numbers,

$$|n_{1,+}, n_{0,+}, n_{-1,+}, n_{1,-}, n_{0,-}, n_{-1,-}\rangle\,. \tag{9}$$

This Hamiltonian has a set of symmetries that will be the key for the adiabatic separation later on. We have the total number of particles

$$\hat{N} = \sum_{\substack{k \in \{-1,0,1\} \\ \sigma \in \{-,+\}}} \hat{n}_{k,\sigma}, \tag{10}$$

and the total angular momentum

$$\hat{L} = \sum_{\substack{k \in \{-1,0,1\} \\ \sigma \in \{-,+\}}} k \cdot \hat{n}_{k,\sigma}. \tag{11}$$

Using (8), it is easy to show that

$$[\hat{H}, \hat{N}] = [\hat{H}, \hat{L}] = 0,\tag{12}$$

and the Hilbert space can be divided into sectors with the respective quantum numbers $(N, L)$. To simplify the task we will focus on the special case of fixed $N$ and $L = 0$. Except for the derivation of the effective Hamiltonian which is done for general $L$. In this way, the effective number of degrees of freedom is reduced from six to four.

Besides these two symmetries, the energy spectrum splits up symmetrically in the positive and negative direction, as can be seen in Figure 1. For an even particle number $N$ this observation can be explained using the operator

$$\hat{\xi} = \otimes_{\alpha=1}^{N} \hat{\sigma}_x^{(\alpha)} (-1)^{\frac{\hat{S}}{2}}\tag{13}$$

where $\hat{S}$ is the total (pseudo) spin

$$\hat{S} = \sum_{\alpha=1}^{N} \hat{\sigma}_z^{(\alpha)}\tag{14}$$

that satisfies

$$(-1)^{\frac{\hat{S}}{2}} \cdot (-1)^{-\frac{\hat{S}}{2}} = 1,\tag{15}$$

and therefore it is easy to show that

$$\langle \psi | \hat{\xi}^\dagger \hat{H} \hat{\xi} | \psi \rangle = - \langle \psi | \hat{H} | \psi \rangle.\tag{16}$$

As $\hat{\xi}$ is a bijection on the set of eigenstates $|\psi\rangle$ of $\hat{H}$ with energy $E = \langle \psi | \hat{H} | \psi \rangle$, there always exists a state $|\phi\rangle = \hat{\xi} |\psi\rangle$ that is also an eigenstate of $\hat{H}$. The energy value corresponding to this state is then given by

$$E_\phi = \langle \phi | \hat{H} | \phi \rangle = -E.\tag{17}$$

Finally, a parity operator $\hat{P}$ can be defined which simultaneously flips all spins and momenta, given by a complex conjugation to invert the momenta in the eigenbasis of plane waves followed by a spin flip,

$$\hat{P} = \otimes_{\alpha=1}^{N} \hat{\sigma}_x^{(\alpha)} (\cdot)^*,\tag{18}$$

satisfying $\hat{P}^2 = 1$. Also, since $[\hat{P}, \hat{H}] = 0$, $\hat{P}$ represents a discrete symmetry that splits the Hilbert space into two separate subspaces leading to a separation of the energy spectrum into two independent subspectra (In general, one does have $[\hat{P}, \hat{L}] \neq 0$; however, for $L = 0$ the two operators commute.)

$$H = \begin{pmatrix} H_+ & 0 \\ 0 & H_- \end{pmatrix},\tag{19}$$

see Figure 1.

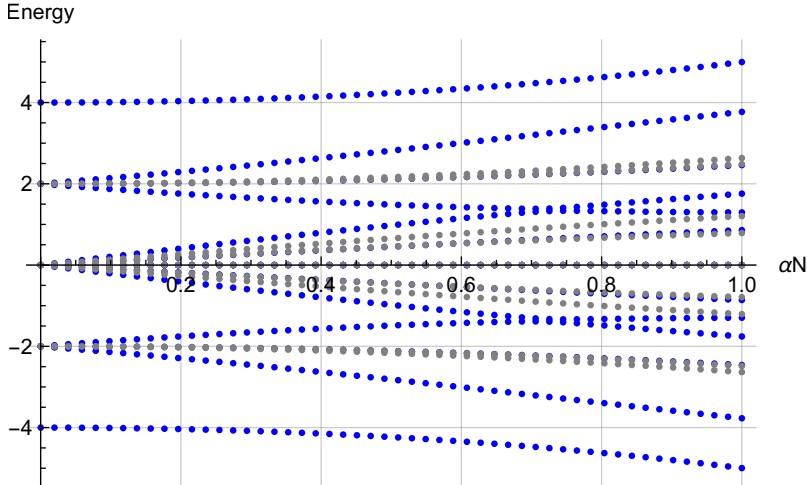

**Figure 1.** Energy spectrum for $N = 4$, $L = 0$ splitted into positive (blue) and negative (gray) parity. Scaled units $[E] = \frac{\hbar}{R}$ used.

As a final remark, we note that the existence of further symmetries is ruled out by a numerical diagonalization and the analysis of avoided crossings, as indicated for $N = 20$, $L = 0$, $P = 1$ in Figure 2. The absence of real crossing suggests that there are no additional symmetries to be found which could be used to further reduce the dimensions of the Hamiltonian (8) [30].

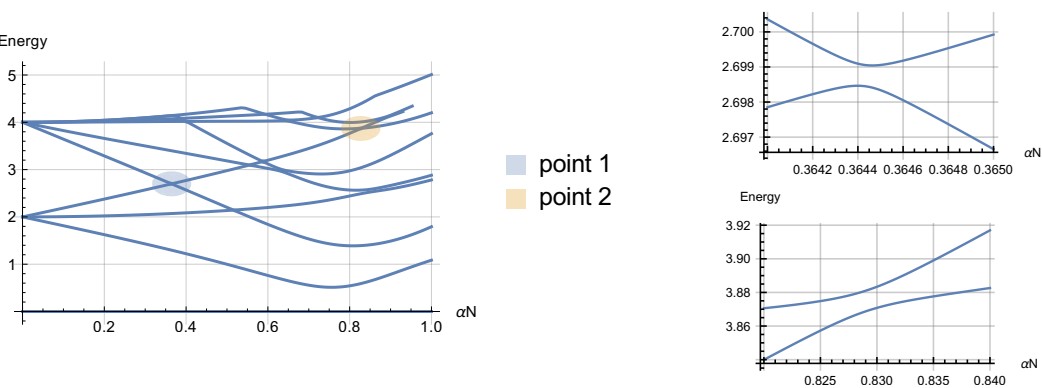

**Figure 2.** Excitation spectrum at $N = 120$, $L = 0$ (**left**) and zooms into two exemplarily points which display avoided crossings (**right**). Scaled units $[E] = \frac{\hbar}{R}$ used.

## 3. Adiabatic Separation of the Hamiltonian

Using

$$n_0 \equiv n_{0,+} + n_{0,-}, \tag{20}$$

which corresponds to the total number of particles in the zero modes, we can rearrange the Fock basis into several blocks. Figure 3 shows the wavefunction of the ground state and the first five excited states of the system for $N = 120$, $L = 0$, $\alpha N = 0.7$. The vertical grid lines indicate the borders between the different blocks of the Fock basis which are arranged in ascending values of $n_0$. Within one block the states are further sorted with respect to $n_{\text{imb}} \equiv n_{0,+} - n_{0,-}$ which characterizes the imbalance between the occupation of the zero modes of a Fock state.

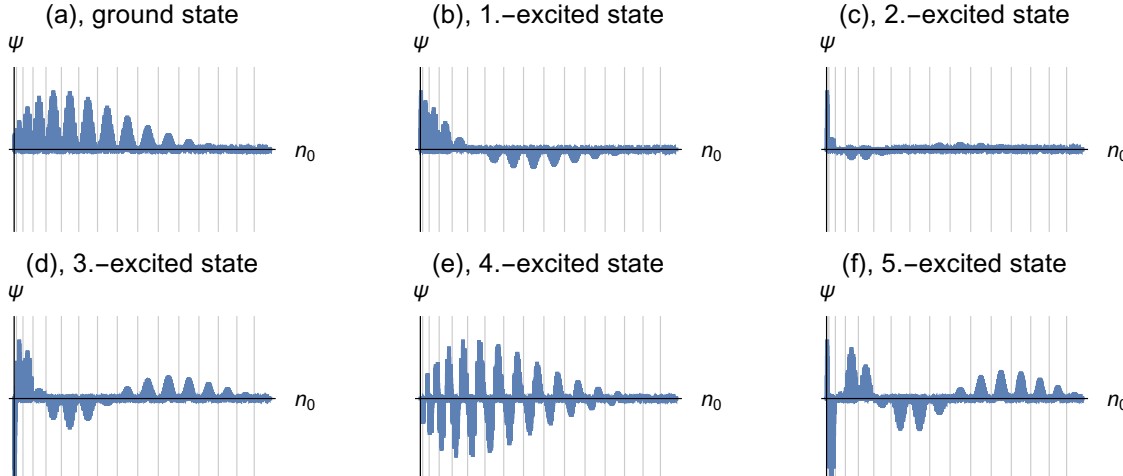

**Figure 3.** The wavefunctions $\psi$ of the six energetically lowest states for $\alpha N = 0.7$, $L = 0$ and $N = 120$. Along the horizontal axis we order the Fock basis for fixed $N$ and $L$ into sectors of constant zero mode occupation $n_0 = n_{0,+} + n_{0,-}$, and within these blocks, we further order the basis according to the imbalance between the zero modes $n_0 = n_{0,+} - n_{0,-}$. The further subordering, using the last two remaining degrees of freedom, is not chosen in a specific way. This representation exhibits particle in a box-type excitations in the sectors of constant $n_0$.

Inspection of the wavefunctions in Figure 3 indicates a further substructure: Within each $n_0$-subspace the wavefunction has a form corresponding to the ground (see panels Figure 3a–d and Figure 3f) or excited (see Figure 3e) state of a particle in a box whereas over the whole Fock space these fine structures are enveloped by an overall oscillation. Going even further, this kind of behaviour can be compared to the excitation spectrum of a molecule in the Born-Oppenheimer approximation [31]. In this picture, the behaviour within a constant $n_0$-subspace corresponds to a fast degree of freedom which separates the energy spectrum into different channels [32]. Within each channel, there are smaller excitations which are determined by the slow degree of freedom corresponding to the behaviour of the oscillations in the envelope.

Based on this physical motivation we now take a look at the matrix representation of the Hamiltonian (8). If we choose $N > 0$ and order the Fock basis in blocks of constant $n_0$ including both parities $P = \pm 1$, one obtains a tridiagonal block matrix

$$H = \begin{pmatrix} H_0 & H_{0,2} & 0 & & & \\ H_{2,0} & H_2 & \ddots & & \ddots & \\ 0 & \ddots & \ddots & \ddots & & 0 \\ & \ddots & \ddots & H_{N-2} & H_{N-2,N} \\ & & 0 & H_{N,N-2} & H_N \end{pmatrix}, \tag{21}$$

where $H_{n_0}$ is the projection of the Hamiltonian (8) into the subspace with fixed $n_0$, while $H_{n_0 \pm 2, n_0}$ couples the $n_0$-block to its next neighbours. Due to the form of the interaction all other blocks vanish. The next step is to define transformations $U_{n_0}$ which diagonalize $H_{n_0}$ and thereby the global transformation

$$U = \begin{pmatrix} U_0 & 0 & & & \\ 0 & U_2 & \ddots & & \\ & \ddots & \ddots & \ddots & \\ & & \ddots & U_{N-2} & 0 \\ & & & 0 & U_N \end{pmatrix} \tag{22}$$

from the Fock space into a basis that diagonalizes each projection of the Hamiltonian (8) to an $n_0$-subspace. This allows us to systematically select vectors solely corresponding to the ground, first or second excited states of the channels and project out all the others. This projection then neglects all possible couplings between different channels. Please note that this procedure has to be repeated for every $\alpha N$ as the magnitude of the interaction alters the corresponding eigenvectors of the $n_0$-blocks.

The resulting spectrum is shown in Figure 4. Neglecting the coupling of different channels is fully justified as seen from the excellent agreement between the exact and approximated spectrum.

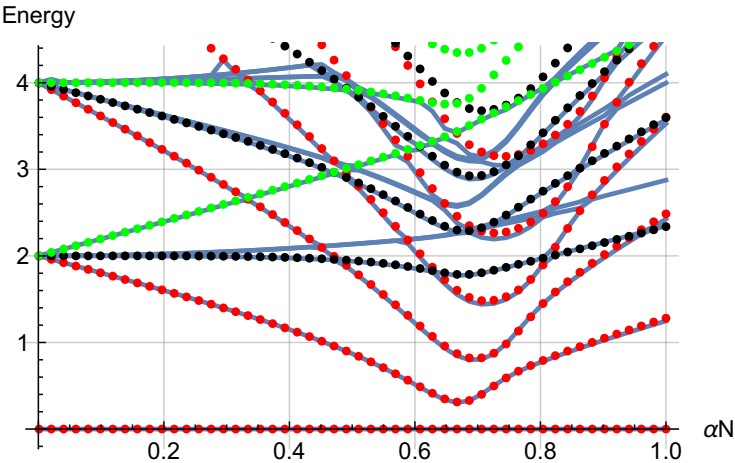

**Figure 4.** Spectrum based on the adiabatic approximation (22) (dots) compared to the exact spectrum (lines) at $N = 70$, $L = 0$. The approximated dots are obtained by restriction to the ground state (red), first (black) and second (green) excited state within the fast degree of freedom. Scaled units $[E] = \frac{\hbar}{R}$ used.

Here, the solid blue lines show the energy levels of the full Hamiltonian (8). In comparison, the dotted lines show the excitation spectrum if the Hamiltonian is restricted to different single channels. They correspond to restrictions to the ground state (red), first (black) and second (green) excited state within the fast degree of freedom. The excellent agreement shows that this approximation provides energy levels of the original system quantitatively to very good accuracy. Furthermore, it enables us to split the spectrum into several subspectra which can be investigated independently of each other.

## 4. Effective Hamiltonian

The subsequent derivation is carried out for chosen particle number $N$ and total momentum $L$, such that the respective operators are replaced by these quantum numbers. To properly derive the block structure of the Hamiltonian (8), an operator $\hat{P}_{n_0}$ can be defined that projects the Hilbert space onto its subspace with constant $n_0$. By definition we have

$$\sum_{n_0=0}^{N} \hat{P}_{n_0} = \hat{1}, \tag{23}$$

which can be used to rewrite the Hamiltonian as

$$\hat{H} = \sum_{n_0,n_0'} \hat{P}_{n_0'} \hat{H} \hat{P}_{n_0} = \sum_{n_0} \hat{H}_{\text{eff}}(n_0) + \sum_{n_0,n_0'} \hat{H}_{\text{coup}}(n_0', n_0), \tag{24}$$

where the second term is the coupling Hamiltonian. The part diagonal in $n_0$ is the effective Hamiltonian

$$\hat{H}_{\text{eff}}(n_0) = \hat{H}_0(n_0) + \hat{H}_1(n_0) + \hat{H}_2(n_0), \tag{25}$$

that defines the adiabatically separated channels. In this division $H_0(n_0)$ represents the kinetic part of the Hamiltonian (8). $H_2(n_0)$ are the parts of the interaction which contain bosonic operators $a_{k,\sigma}, \hat{a}_{k,\sigma}$ with $k \in \{\pm 1\}$ and commute with $\hat{n}_0$. Last of all in $H_1(n_0)$ we summed up all the remaining parts of the interaction, which commute with $\hat{n}_0$. This $H_{\text{eff}}$ is the starting point of all further analysis.

*4.1. Redefinition of Zero Modes*

The effective Hamiltonian (25) has an additional constant of motion,

$$\hat{F} \equiv \sum_{\sigma=\pm 1} \hat{a}_{0,\sigma}^{\dagger} \hat{a}_{0,-\sigma} \tag{26}$$

that can be easily shown to commute also with $\hat{N}$ and $\hat{L}$. The redefinition of the creation and annihilation operators of the zero modes

$$\hat{z}_{\pm} \equiv \frac{1}{\sqrt{2}} (\hat{a}_{0,+} \pm \hat{a}_{0,-}) \tag{27}$$

gives

$$\sum_{\sigma=\pm 1} \hat{a}_{0,\sigma}^{\dagger} \hat{a}_{0,\sigma} \to \hat{z}_+^{\dagger} \hat{z}_+ - \hat{z}_-^{\dagger} \hat{z}_-, \tag{28}$$

while $\hat{n}_0 = \hat{z}_+^{\dagger} \hat{z}_+ + \hat{z}_-^{\dagger} \hat{z}_-$ keeps its structure. A further definition offers a new good quantum number necessary to describe the effective system:

$$\hat{c}_0 = \hat{z}_+^{\dagger} \hat{z}_+, \quad [\hat{H}_{\text{eff}}(n_0), \hat{c}_0] = 0. \tag{29}$$

Therefore we are able to rewrite $\hat{H}_1(n_0)$ in diagonal form as

$$\hat{H}_1(n_0) = \frac{\alpha}{2} (2N - n_0 - 2)(2\hat{c}_0 - n_0), \tag{30}$$

where the range of this new quantum quantum number $c_0 \in \{0, 1, \dots, n_0\}$ depends on $n_0$. Please note that the operator $\hat{c}_0$ is deliberately choosen in a way such that the resulting eigenenergy

$$E_1(n_0, c_0) = \frac{\alpha}{2}(2N - n_0 - 2)(2c_0 - n_0) \tag{31}$$

of $\hat{H}_1(n_0)$ is minimal for $c_0 = 0$.

*4.2. Redefinition of Kinetic Modes*

Now we focus on the remaining parts of the effective Hamiltonian (25) to show how it can be rendered diagonal by a redefinition of the creation and annihilation operators. Up to now $\hat{H}_{\text{eff}}(n_0)$ (25) consists of two parts. While the first one ($E_1(n_0, c_0)$) was analyzed in Section 4.1, the second part looks comparatively difficult:

$$\hat{H}_0(n_0) + \hat{H}_2(n_0) = \sum_{\substack{k \in \{-1,1\} \\ \sigma \in \{-,+\}}} \sigma k \cdot \hat{a}_{k,\sigma}^{\dagger} \hat{a}_{k,\sigma} + \sum_{k=\pm 1} -\frac{\alpha}{4}(3N + n_0 + k \cdot L - 2)\hat{h}_k, \tag{32}$$

where $\hat{h}_k$ is defined as

$$\hat{h}_k \equiv \hat{a}_{k,+}^{\dagger} \hat{a}_{k,-} + \hat{a}_{k,-}^{\dagger} \hat{a}_{k,+}, k \in \{\pm 1\}. \tag{33}$$

It is quadratic in the creation and annihilation operators

$$\hat{a}_{k,+}^{\dagger} \hat{a}_{k,-}, k \in \{\pm 1\}, \tag{34}$$

suggesting to define a vector

$$v \equiv \begin{pmatrix} \hat{a}_{1,+} & \hat{a}_{1,-} & \hat{a}_{-1,+} & \hat{a}_{-1,-} \end{pmatrix}^{\mathrm{T}}, \tag{35}$$

containing all annihilation operators of the $(k = \pm 1)$-modes, that allows us to rewrite the Hamiltonian (32) as

$$\hat{H}_0 + \hat{H}_2 = v^{\dagger} M v, \qquad\qquad M \equiv \begin{pmatrix} A_+ & 0 \\ 0 & A_- \end{pmatrix}, \tag{36}$$

where

$$A_+ \equiv \begin{pmatrix} 1 & -\frac{\alpha}{4}(K-L) \\ -\frac{\alpha}{4}(K-L) & -1 \end{pmatrix}, \qquad A_- \equiv \begin{pmatrix} -1 & -\frac{\alpha}{4}(K+L) \\ -\frac{\alpha}{4}(K+L) & 1 \end{pmatrix}, \tag{37}$$

and $K > 0$ depends on $N$ and $n_0$ via

$$K \equiv 3N + n_0 - 2. \tag{38}$$

The quadratic form (36) allows us to diagonalize the Hamiltonian (32). From the blockstructure of the matrix one can already conclude that the diagonalization will only mix those operators within the same k-mode,

$$\begin{pmatrix} \hat{p}_+ \\ \hat{p}_- \end{pmatrix} \equiv C_+ \begin{pmatrix} \hat{a}_{1,+} \\ \hat{a}_{1,-} \end{pmatrix}, \qquad\qquad \begin{pmatrix} \hat{n}_+ \\ \hat{n}_- \end{pmatrix} \equiv C_- \begin{pmatrix} \hat{a}_{-1,+} \\ \hat{a}_{-1,-} \end{pmatrix}, \tag{39}$$

where $C_{\pm}$ are matrices obtained from the eigenvectors of $A_{\pm}$. This notation is chosen in such a way that "$p$" corresponds to the new operators obtained from the operators acting on "positive" k-modes and "$n$" from the "negative" ones. Furthermore, the "+", "−" indices (not to be confused with the eigenvalues of the parity operator) of the new operators refer to the associated eigenvalues of the diagonalized matrix

$$C_{\pm} A_{\pm} C_{\pm}^T = \begin{pmatrix} \sqrt{1 + (\frac{\alpha}{4}(K \mp L))^2} & 0 \\ 0 & -\sqrt{1 + (\frac{\alpha}{4}(K \mp L))^2} \end{pmatrix}. \tag{40}$$

As this redefinition is a rotation of the old operators, the sum of their occupation numbers remains unaffected,

$$\hat{n}_1 \equiv \hat{n}_{1,+} + \hat{n}_{1,-} = \hat{a}_{1,+}^{\dagger}\hat{a}_{1,+} + \hat{a}_{1,-}^{\dagger}\hat{a}_{1,-} = \hat{p}_+^{\dagger}\hat{p}_+ + \hat{p}_-^{\dagger}\hat{p}_-, \tag{41}$$

and the same holds true for the negative k-modes

$$\hat{n}_{-1} \equiv \hat{n}_{-1,+} + \hat{n}_{-1,-} = \hat{a}_{-1,+}^{\dagger}\hat{a}_{-1,+} + \hat{a}_{-1,-}^{\dagger}\hat{a}_{-1,-} = \hat{n}_+^{\dagger}\hat{n}_+ + \hat{n}_-^{\dagger}\hat{n}_-. \tag{42}$$

Finally, in view of the transformation from Section 4.1, we are able to fully diagonalize the effective Hamiltonian (25)

$$\hat{H}_{\mathrm{eff}}(n_0) = \frac{\alpha}{2}(2N - n_0 - 1)(2\hat{c}_0 - n_0) + \sqrt{1 + \left(\frac{\alpha}{4}(K-L)\right)^2} \cdot (\hat{p}_+^{\dagger}\hat{p}_+ - \hat{p}_-^{\dagger}\hat{p}_-)$$

$$+ \sqrt{1 + \left(\frac{\alpha}{4}(K+L)\right)^2} \cdot (\hat{n}_+^{\dagger}\hat{n}_+ - \hat{n}_-^{\dagger}\hat{n}_-). \tag{43}$$

This can be made explicit using the eigenbasis of the operators

$$\hat{c}_+ \equiv \hat{p}_+^\dagger \hat{p}_+, \qquad\qquad \hat{c}_- \equiv \hat{n}_+^\dagger \hat{n}_+, \qquad (44)$$

that commute with $\hat{H}_{\text{eff}}(n_0)$. Using

$$L = n_1 - n_{-1}, \qquad\qquad N - n_0 = n_1 + n_{-1}, \qquad (45)$$

one gets the explicit expression

$$\begin{aligned}
E_{\text{eff}}(n_0, c_0, c_+, c_-) = \frac{\alpha}{2}(2N - n_0 - 1)(2c_0 - n_0) &+ \sqrt{1 + \left(\frac{\alpha}{4}(K - L)\right)^2} \cdot \left(2c_+ - \frac{N - n_0 + L}{2}\right) \\
&+ \sqrt{1 + \left(\frac{\alpha}{4}(K + L)\right)^2} \cdot \left(2c_- - \frac{N - n_0 - L}{2}\right)
\end{aligned} \qquad (46)$$

for the eigenenergies. Please note that the range of the new quantum numbers

$$c_\pm \in \left\{0, 1, \ldots, \frac{N - n_0 \pm L}{2}\right\} \qquad (47)$$

is defined by $N$, $L$ and $n_0$, while in the case of $L = 0$, (46) simplifies to

$$E_{\text{eff}}(n_0, c_0, c_+, c_-) = \frac{\alpha}{2}(2N - n_0 - 1)(2c_0 - n_0) + \sqrt{1 + \left(\frac{\alpha}{4}K\right)^2} \cdot (2(c_+ + c_-) - (N - n_0)). \qquad (48)$$

Each combination of quantum numbers $(c_0, c_+, c_-)$ then defines a different channel within the effective Hamiltonian (43). In a last step, we assume that interactions between different channels can be neglected as motivated in Section 3. Within an $(c_0, c_+, c_-)$-channel this leaves only one possible combination

$$\hat{p}_-^\dagger \hat{n}_-^\dagger \hat{z}_- \hat{z}_- \qquad (49)$$

and its Hermitian conjugate, leading to an approximated single-channel Hamiltonian

$$\hat{H}_{\text{approx}}(c_0, c_+, c_-) = E_{\text{eff}}(\hat{n}_0, c_0, c_+, c_-) - \frac{\alpha}{2}\left[\left(1 + \frac{\hat{a}}{\sqrt{1 + \hat{a}^2}}\right)\hat{p}_-^\dagger \hat{n}_-^\dagger \hat{z}_- \hat{z}_- + h.c.\right] \qquad (50)$$

$$\text{with } \hat{a} \equiv \frac{\alpha}{4}(3N + \hat{n}_0 - 2)$$

for the channel labeled by $(c_0, c_+, c_-)$.

Figure 5 presents the decoupled energy spectrum of this system resulting from the Hamiltonian (50). The contribution of $c_+$ and $c_-$ to the approximate energy $E_{\text{eff}}(n_0, c_0, c_+, c_-)$ depends only on their sum $c_+ + c_-$ for the case $L = 0$, and therefore $c_-$ was chosen to be always zero. The resulting spectrum is plotted (marked by dots) against the one (solid lines) from the complete Hamiltonian (8). While the higher excitations show small deviations, the results are essentially the same as without the approximation. In particular, as the main interest is in the lowest channel which corresponds to the black dots, the new quantum numbers give rise to the ability to split the spectrum into several combinations of $\{c_0, c_+, c_-\}$. Further investigations will be focused on the ground state and the lowest excitations which means that these quantum numbers are always chosen to be zero.

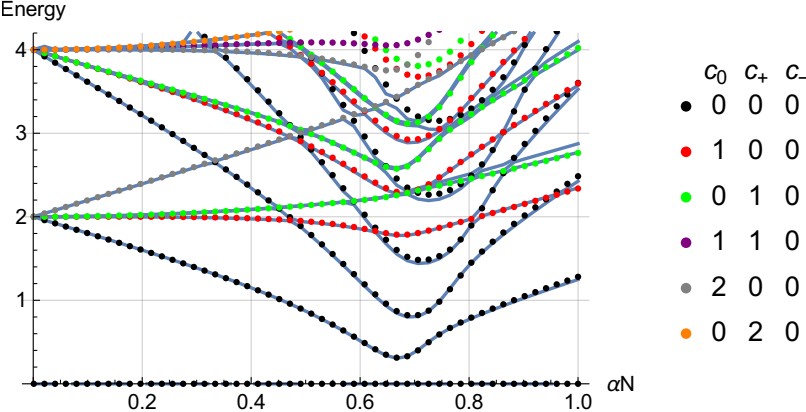

**Figure 5.** The exact energy spectrum (solid) evaluated for $N = 70$, $L = 0$. Above it is plotted the decoupled energy spectrum (dots), obtained by numerical diagonalization of (50), with the corresponding effective quantum numbers. In choice of the parameters and approximations this is equivalent to Figure 4, but after the previous derivation the division into several channels is now structured by the effective quantum numbers. Scaled units $[E] = \frac{\hbar}{R}$ used.

## 5. Classical Analysis

In the following, we will analyse the critical properties of our effective model (50) by means of a semiclassical analysis. Starting with the diagonal part (43)

$$
\begin{aligned}
\hat{H}_{\text{eff}}(n_0) = &\frac{\alpha}{2}(2N - n_0 - 1)(\hat{z}_+^\dagger \hat{z}_+ - \hat{z}_-^\dagger \hat{z}_-) \\
&+ \sqrt{1 + \left(\frac{\alpha}{4}(3N + n_0 - 2)\right)^2} \cdot (\hat{p}_+^\dagger \hat{p}_+ - \hat{p}_-^\dagger \hat{p}_- + \hat{n}_+^\dagger \hat{n}_+ - \hat{n}_-^\dagger \hat{n}_-),
\end{aligned}
\tag{51}
$$

we substitute the creation/annhihlation operators by classical phase space variables

$$
\hat{f}_\sigma \to \sqrt{n_{f,\sigma}} \cdot e^{i\phi_{f,\sigma}}, \quad f \in \{z, p, n\}, \quad \sigma \in \{+, -\},
\tag{52}
$$

and neglect all terms of order $O(N^0)$ in the limit of $N \to \infty$, to obtain

$$
\begin{aligned}
E_{\text{eff,cl}} &= \frac{\alpha}{2}(2N - n_0)(n_{z,+} - n_{z,-}) + \cosh(\gamma)(n_{p,+} - n_{p,-} + n_{n,+} - n_{n,-}), \\
\sinh(\gamma(\alpha, N, n_0)) &\equiv \frac{\alpha}{4}(3N + n_0)
\end{aligned}
\tag{53}
$$

where "cl" refers to the classical (mean field) limit. Since the coupling between different channels can be neglected, as shown in Sections 3 and 4, the classical form of the remaining interaction then gives

$$
H_{\text{coup,cl}}(n_0) = \frac{\alpha}{2}\left(1 + \tanh(\gamma)\right) n_{z,-} \sqrt{n_{p,-} n_{n,-}} \cdot \cos(2\phi_{z,-} - \phi_{p,-} - \phi_{n,-}).
\tag{54}
$$

To get an easily solvable form we reduce the Hamiltonian (51) to its channel of minimal energy by setting

$$
n_{z,+} = n_{p,+} = n_{n,+} = 0,
\tag{55}
$$

while we reexpress $\{n_{z,-}, n_{p,-}, n_{n,-}\}$ in terms of $N, L$ and $n_0$ through the point transformation

$$n_0 = n_{z,-}, \qquad\qquad n_{p,-} = n_{n,-} = \frac{N - n_0}{2},$$

$$\theta = \phi_{z,-} - \frac{1}{2}(\phi_{p,-} + \phi_{n,-}), \qquad \theta_N = \frac{1}{2}(\phi_{p,-} + \phi_{n,-}), \quad \theta_L = \frac{1}{2}(\phi_{p,-} - \phi_{n,-}). \tag{56}$$

This finally leads to a one-dimensional description with only two (conjugate) phase-space coordinates $n_0$ and $\theta$,

$$E_{\text{cl}}(\alpha, \phi, z) = -\cosh(\gamma)(N - n_0) - \frac{\alpha}{2} n_0 \left((2N - n_0) + (N - n_0)(1 + \tanh(\gamma))\cos(2\theta)\right). \tag{57}$$

To extract the physical properties of this mean field Hamiltonian (57), valid for $\lim N \to \infty$, we define scaled variables

$$e_{\text{cl}} = \frac{E_{\text{cl}}}{N}, \qquad z = \frac{n_0}{N} \in [0,1], \qquad \bar{\alpha} = \alpha N, \qquad \sinh(\gamma) = \sinh(\gamma(\bar{\alpha},z)) = \frac{\bar{\alpha}}{4}(3 + z) \tag{58}$$

to get the energy per particle as

$$e_{\text{cl}}(\bar{\alpha}, \theta, z) = -\cosh(\gamma(\bar{\alpha},z))(1 - z) - \frac{\bar{\alpha}}{2} z \left((2 - z) + (1 - z)(1 + \tanh(\gamma(\bar{\alpha},z)))\cos(2\theta)\right). \tag{59}$$

We are now ready to proceed with the study of the classical phase space. Obviously, it is $\pi$-periodic in $\theta$ such that the analysis can be restricted to $\theta \in [-\frac{\pi}{2}, \frac{\pi}{2}]$.

Figure 6 shows contour plots of the energy $e_{\text{cl}}$ for different values of the coupling $\bar{\alpha}$. As clearly seen, there is a qualitative change within the phase space, when the scaled interaction is increased from $\bar{\alpha} = 0$ and $\bar{\alpha} = 1$. While in the non-interacting case, the phase space allows only rotations (Using the analogy to the mathematical pendulum), the phase space is divided into two qualitatively different regions at $\bar{\alpha} = 1$. The regime of the lowest energies consists of vibrations/librations, separated from the rotating orbits by a separatrix. This separatrix is created at $z = \phi = 0$ at a critical interaction $\alpha_{\text{crit}}$. Furthermore, for weak interaction ($0 \leq \bar{\alpha} \leq \bar{\alpha}_{\text{crit}}$) the energy minimum is located at $z = 0$ and degenerate in $\theta$. In contrast, at a stronger interaction ($\bar{\alpha}_{\text{crit}} < \bar{\alpha}$), the energy minimum consists of only one discrete point $z > 0$, $\theta = 0$.

According to its definition, $z$ represents the ratio of particles within the zero modes $n_{0,+}$ and $n_{0,-}$ with respect to the whole particle number $N$. This yields the interpretation that, for an interaction greater than $\bar{\alpha}_{\text{crit}}$, the occupation of the zero modes within the ground state changes from a microscopic occupation near zero to a macroscopic one at a finite value and therefore indicates that $z$ can be taken as an order parameter characterizing a type of quantum phase transition. Since in this model there is no physical ground state, the abrupt changes that characterize quantum phase transition happen now around an effective, pseudo-ground state, and properly speaking we should refer to this as a pseudo-quantum phase transition. In the spirit of not overcharging with new terminology the manuscript, however, we used and continue using the term quantum phase transition along the text. This is in complete analogy with the spin-one Bose gas without pseudospin and quadratic Zeeman shift [33] and the truncated versions of the attractive one-dimensional Bose gas [21].

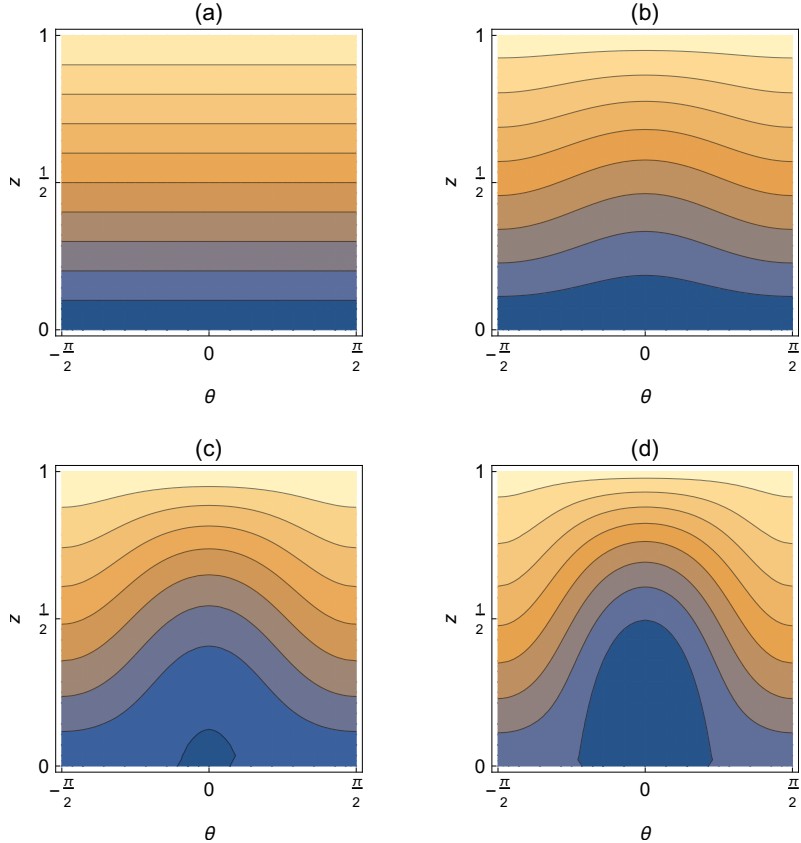

**Figure 6.** Phase diagram of $e_{cl}$ for $\bar{\alpha} = 0$ (**a**), $\frac{1}{3}$ (**b**), $\frac{2}{3}$ (**c**), 1 (**d**). $z = \frac{n_0}{N}$ is the normalized zero mode occupation and $\theta$ the conjugate phase.. The color scaling describes the value of the energy. Blue represents the minimum and light orange the maximum.

## 6. Analytic Analysis of the Quantum Phase Transition

Armed with a clear signature of a phase transition in the change of morphology of the classical (mean field) limit produced by the appearance of the separatrix, we will now study the different aspects of this critical behaviour. As discussed before in Section 5, the energy minimum is always located at $\theta = 0$ for $\bar{\alpha} > \bar{\alpha}_{crit}$ and is degenerate in $\theta$ for $\bar{\alpha} \leq \bar{\alpha}_{crit}$. Therefore, this variable can be eliminated in the following discussion by setting $\theta = 0$. The resulting energy dependence $e_{cl}(\bar{\alpha}, \theta = 0, z)$ on $z$ for several $\bar{\alpha}$ is shown in Figure 7.

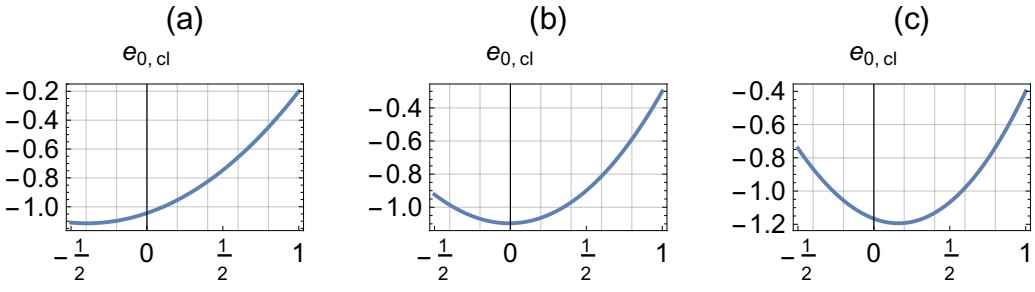

**Figure 7.** $e_{cl}(\bar{\alpha}, \theta = 0, z)$ for $\alpha N = 0.4$ (**a**), 0.6 (**b**), 0.8 (**c**).

The range of $z$ was deliberately chosen as $\{-\frac{1}{2}, 1\}$, despite the fact that negative $z$ are unphysical according to its definition, to illustrate the behaviour of the local minimum depending on the interaction strength $\bar{\alpha}$. For $\bar{\alpha} \leq \bar{\alpha}_{crit}$ this minimum would be at $z^* < 0$. As this is not part of the allowed phase

space, the minimum will simply be located at $z^* = 0$. If the interaction strength is increased, $z^*$ increases too until it reaches $z^* = 0$. This is exactly the point where the quantum phase transition can be expected. To find the critical value, one cane use that the derivative of the energy $e_{cl}$ with respect to $z$ should vanish when evaluated at $z = 0$ and $\bar{\alpha} = \bar{\alpha}_{crit}$

$$\left. \frac{\partial e_{cl}(\bar{\alpha}_{crit}, \theta = 0, z)}{\partial z} \right|_{z=0} = -\frac{3\bar{\alpha}_{crit}}{2} + \frac{4}{\sqrt{16 + 9\bar{\alpha}_{crit}^2}} = 0, \tag{60}$$

which provides the critical parameter as

$$\bar{\alpha}_{crit} = \frac{2}{3}\sqrt{2(\sqrt{2} - 1)} \approx 0.607. \tag{61}$$

To further prove this critical behaviour, Figure 8 shows the functional dependence of the second derivative of energy minimum with respect to $\bar{\alpha}$,

$$\frac{\partial^2 e_{cl}(\bar{\alpha}, \theta = 0, z_{min})}{\partial^2 \bar{\alpha}}, \quad \text{with} \quad e_{cl}(\bar{\alpha}, \theta = 0, z_{min}) \equiv e_{cl,min}(\bar{\alpha}). \tag{62}$$

The plot consists of four curves and a dashed line indicating the exact $N \to \infty$ values of the discontinuity. All the curves are based on values of the groundstate energy for discrete sets of points of $\bar{\alpha}$, with the second derivative evaluated numerically. The blue dots were calculated using the energy dependence given by the classical Hamiltonian (59), whose minimal energy was numerically determined within the phase space for different values of $\bar{\alpha}$. They are compared to the quantum mechanical results for the ground state at various particle numbers $N$ given by the lowest eigenvalue of the matrix representation of (50) renormalized by $\frac{1}{N}$.

The analytical result $e_{0,<}''(\bar{\alpha})$, is obtained through a simple derivative of the classical energy with respect to $z$ at the critical point

$$e_{0,<}''(\bar{\alpha}_{crit}) = \left. \frac{\partial^2 e_{cl}(\bar{\alpha}_{crit}, \theta = 0, z)}{\partial^2 z} \right|_{z=0} = -\frac{9}{4\sqrt{2}\left(1 + \sqrt{2}\right)^{3/2}} \approx -0.424. \tag{63}$$

Extracting the second value right behind the critical threshold is a bit harder, as the change of the $z$-position depending on $\bar{\alpha}$ has to be taken into account. To this end a leading-order expansion in $z$ is necessary

$$z(\bar{\alpha}) = z(\bar{\alpha}_{crit}) + \left. \frac{\partial z}{\partial \bar{\alpha}} \right|_{\bar{\alpha} = \bar{\alpha}_{crit}} (\bar{\alpha} - \bar{\alpha}_{crit}) + O((\bar{\alpha} - \bar{\alpha}_{crit})^2) \approx z'(\bar{\alpha}_{crit})(\bar{\alpha} - \bar{\alpha}_{crit}), \tag{64}$$

where we used $z(\bar{\alpha}_{crit}) = 0$.

Now problem is reduced to calculating the derivative of $z$ with respect to $\bar{\alpha}$ at the critical point. For this purpose we define the function

$$g(\bar{\alpha}, z) = \frac{\partial e(\bar{\alpha}, \theta = 0, z)}{\partial z}. \tag{65}$$

The zero of this function for a chosen $\bar{\alpha}$ gives the $z$-position of the energy minimum and therefore its derivative is

$$\left. \frac{\partial z}{\partial \bar{\alpha}} \right|_{\bar{\alpha} = \bar{\alpha}_{crit}} = -\left( \frac{\partial g}{\partial z} \right)^{-1}_{\bar{\alpha}_{crit}, z(\bar{\alpha}_{crit}) = 0} \left( \frac{\partial g}{\partial \bar{\alpha}} \right)_{\bar{\alpha}_{crit}, z(\bar{\alpha}_{crit}) = 0}. \tag{66}$$

The last step is to insert $\bar{\alpha}$ into

$$e_{\text{cl}}(\bar{\alpha}, \theta = 0, z) \rightarrow e_{\text{cl}}(\bar{\alpha}, \theta = 0, z(\bar{\alpha}) = z'(\bar{\alpha}_{\text{crit}})(\bar{\alpha} - \bar{\alpha}_{\text{crit}})) \tag{67}$$

and to calculate the second derivative

$$\frac{\partial^2 e_{\text{cl}}(\bar{\alpha}, \theta = 0, z(\bar{\alpha}))}{\partial^2 \bar{\alpha}} = -\frac{9}{1156}\sqrt{\frac{373469}{\sqrt{2}} - \frac{325591}{2}} \approx -2.478. \tag{68}$$

Clearly, the dependence of the ground state energy is seen to be discontinuous at $\bar{\alpha} = \bar{\alpha}_{\text{crit}}$ with $\bar{\alpha}_{\text{crit}}$ determined in the previous section. With the last results we even obtained an analytic expression to quantify the magnitude of the discontinuity

$$e''_{0,<} - e''_{0,>} = \frac{81}{289}\sqrt{569\sqrt{2} - 751} \approx 2.05, \tag{69}$$

in excellent agreement with the numerical result shown in Figure 8.

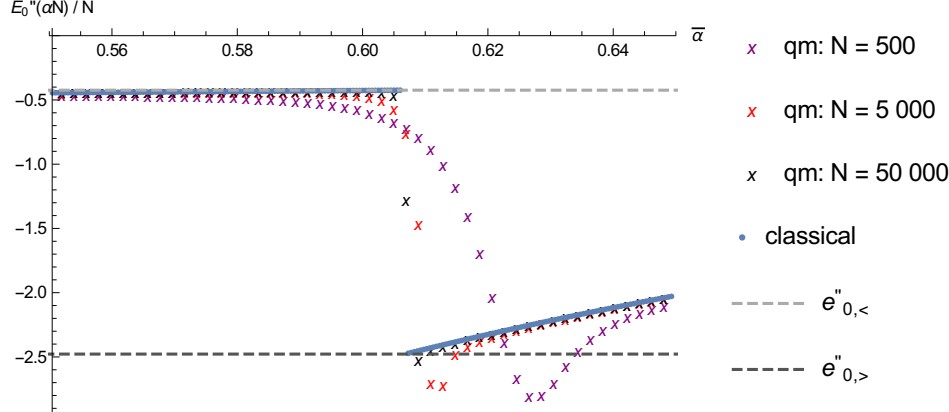

**Figure 8.** Second derivative of the ground state energy with respect to $\bar{\alpha}$. Scaled units $[E] = \frac{\hbar}{R}$ used.

## 7. Further Characterization of the Critical Behaviour

In this last section, we will further characterize the finite-size effects in the quantum phase transition by means of the way the critical parameters approach their sharp values in the mean field limit $N \rightarrow \infty$. Our choice of the appropriate observables comes from the behaviour of the spectrum when we approach the critical region. As seen in Figure 9, and in accordance with what happens in the attractive Lieb-Liniger model [21], one observes a strong accumulation of excited states around criticality, a phenomenon that can be related to an excited-state quantum phase transition [34,35].

The structure of the spectrum in Figure 9 and the dependence shown in Figure 10 suggests that the approach to criticality is well captured by two parameters, namely the minimal gap and interaction value describing its position,

$$\lim_{N \rightarrow \infty} \Delta E_{\text{gap}} = 0, \qquad\qquad \lim_{N \rightarrow \infty} (\bar{\alpha}_{\text{gap}} - \bar{\alpha}_{\text{crit}}) = 0, \tag{70}$$

in the form of a power laws

$$\Delta E_{\text{gap}} \propto N^{-\beta}, \qquad\qquad \Delta\bar{\alpha}_{\text{gap}} \equiv \bar{\alpha}_{\text{gap}} - \bar{\alpha}_{\text{crit}} \propto N^{-\gamma}, \tag{71}$$

where $\beta, \gamma > 0$, will be referred to as dynamical exponents [26].

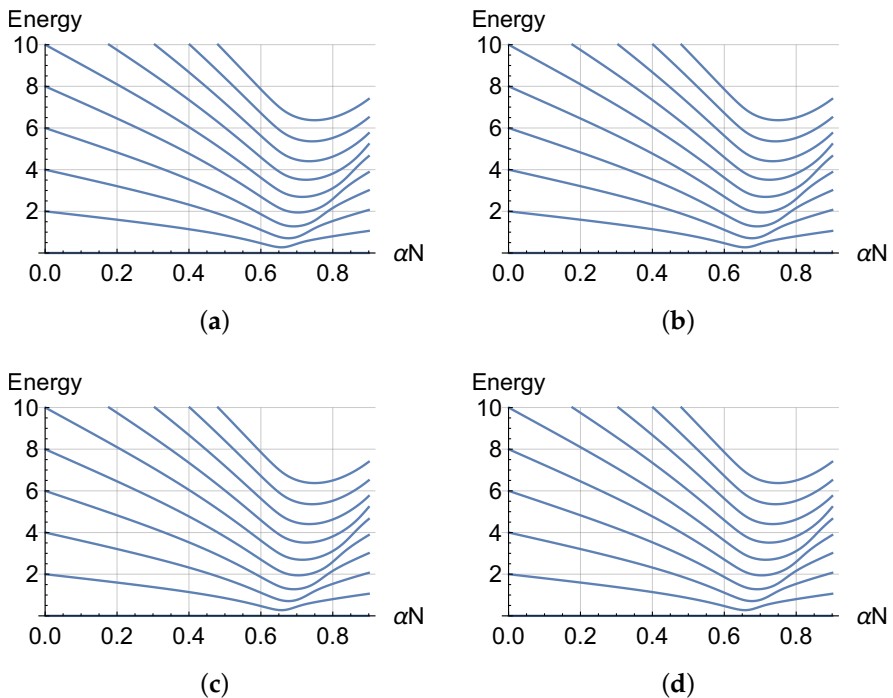

**Figure 9.** Illustration of convergence of the ten lowest energylevels in the first channel towards the critical point for $N = 100$ (**a**), 500 (**b**), 1000 (**c**), 5000 (**d**). Scaled units $[E] = \frac{\hbar}{R}$ used.

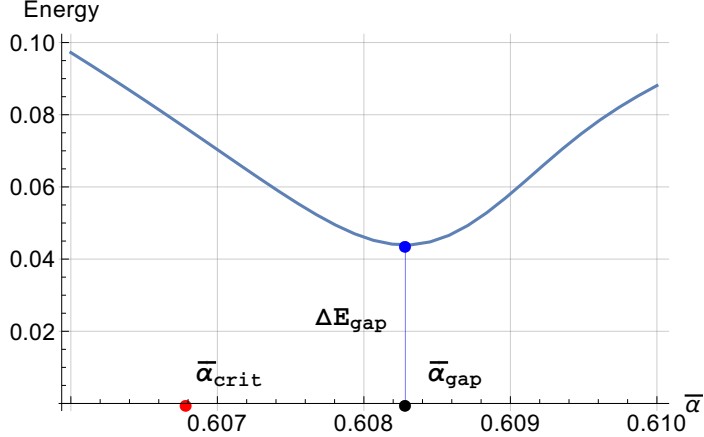

**Figure 10.** Energy gap $\Delta E_{\text{gap}}$ for $N = 20\,000$ with $\bar{\alpha}_{\min} = 0.606$ and $\bar{\alpha}_{\max} = 0.610$. Scaled units $[E] = \frac{\hbar}{R}$ used.

To this end the gap is numerically calculated in a small region between specifically chosen $\bar{\alpha}_{\min}, \bar{\alpha}_{\max}$ for a given particle number $N$. Afterwards, an interpolation function is calculated within this region and the minimum of it is numerically determined. This procedure is repeated for several $N$. Because of its special behaviour at the phase transition, the necessary numerical effort can be reduced drastically [36]. By means of this numerical approach, we are able to present results with particle numbers between twenty and five million. The results are shown in Figure 11 using a double logarithmic scale.

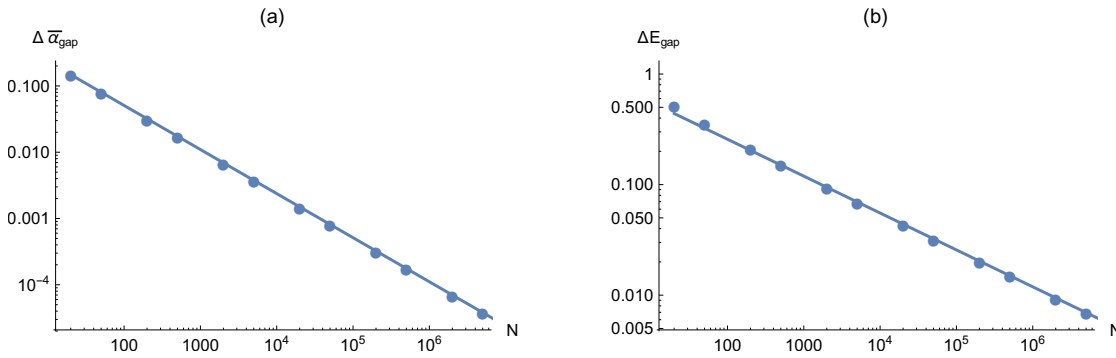

**Figure 11.** Asymptotic behaviour of the gap in interaction $\bar{\alpha}$ (**a**) and energy (**b**) depending on the particle number $N$ in a double logarithmic plot with linear fits.

To extract the power law the particle numbers with $N \geq 20{,}000$ are fitted linearly. Smaller particle numbers are taken out of the fit because this power-law is found to be valid only for large particle numbers. The obtained relations are

$$\Delta\bar{\alpha}_{\text{gap}} \propto N^{-0.3336}, \qquad \Delta E_{\text{gap}} \propto N^{-0.6651}, \qquad (72)$$

where the powers seem to coincide with the values $-\frac{1}{3}$ and $-\frac{2}{3}$ within small tolerance.

These scalings rule how the mean-field limit $N \to \infty$ is approached by two purely quantum observables, as by their very definition the minimal gap and corresponding critical interaction require quantization. An analytical approach that allows for a physical picture and the prediction of such scaling exponents lies therefore beyond the realm of the mean-field approach. A proper semiclassical analysis, able to study such effects by quantizing the mean-field phase space, was successfully applied for the non-relativistic, spinless case in [21,28], where the dynamical exponents were found to be exactly given by $\frac{1}{3}$ and $\frac{2}{3}$. We expect that an extension of the semiclassical quantization of [21,28] in the present relativistic case is feasible given the close similarities in the effective phase space, and the study of the corresponding scaling laws is work in progress.

## 8. Summary and Conclusions

In this article, we explored a (pseudo) relativistic extension of the attractive Lieb-Liniger model, by considering both particles with linear dispersion and spin degree of freedom. Our objective was to check the existence of a relativistic analogue of the well-known quantum phase transition [26] displayed by the original non-relativistic model, where the attractive potential drives a transition of the ground state from a homogeneous state into an inhomogeneous one due to the critical appearance of a bright soliton, as thoroughly study by means of semiclassical methods in [21].

As a main result, we find numerically and explain analytically that the relativistic extension indeed shows clear signatures of critical behaviour and a quantum phase transition where the macroscopic occupation of the side modes ($|k| = 1$), characterized by the vanishing order parameter given by the occupation of the homogeneous zero modes, is destroyed by quantum fluctuations giving rise to the macroscopic occupation of the zero modes, indicating a sudden broadening of the particle distribution and an increase in the interaction energy.

Given the fact that the existence of the phase transition in the non-relativistic case is essentially due to the quantum integrability of the model, the fact that the same effect can be seen in the present non-integrable system points towards universal aspects of this transition.

To get an analytical understanding of this transition and its connection to the integrability of the non-relativistic case, we followed a combined approach. First, extensive numerical simulations show an adiabatic separation that mimics integrability in the low-energy region. Second, a classical analysis based on this approximate separability of the model allows for understanding the critical behaviour

as a consequence of the appearance of separatrix motion in the mean field limit. This combination enabled us to provide analytical results for the location and characteristics of the quantum phase transition in excellent agreement with exact diagonalization results.

Our work follows the idea of a universal connection between the characteristics of separatrix dynamics in the mean field limit and the parameters describing ground and excited state quantum phase transitions of the quantum system, a subject of particular interest in the field of many-body semiclassics.

**Author Contributions:** B.G. and K.R. devised the project, M.N. and B.G. were the main contributor to the work and M.N. performed the numerical simulations. J.-D.U. devised the manuscript and all the figures were produced by M.N. All authors have read and agreed to the published version of the manuscript.

**Funding:** B.G acknowledges financial support from the Deutsche Forschungsgemeinschaft (DFG) throgh Project No. Ri681/14-1.

**Acknowledgments:** All authors acknowledge discussions with Quirin Hummel. We also acknowledge a very careful reading of the manuscript by an anonymous referee, whose suggestions helped with both content and form of the paper.

**Conflicts of Interest:** The authors declare no conflict of interest.

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
