# Peer review of "Classical and Quantum Signatures of Quantum Phase Transitions in a (Pseudo) Relativistic Many-Body System"

_condensedmatter, doi:10.3390/condmat5020026_

Round 1

Reviewer 1 Report

The authors are investigating a variant of the Lieb-Liniger model using a mean field approach in order to describe the phase transition around an effective ground state and in its excitation spectrum. The current model under investigation is a pseudo-relativistic model, i.e. the equation for the time evolution is the Dirac equation for a 2-dimensional spinor instead of the Schrodinger equation for a scalar wave function as in Lieb-Liniger.
They are able to perform a fully analytical description of the phase transition via a mean-field approach (see below), which is corroborated by extensive numerics.
In Section 2 they list the symmetries of the Hamiltoninan in the truncated Fock space.
In Section 3 they perform an adiabatic approximation. In Section 4 they perform some changes of variable within the adiabiatic approximation to finally diagonalize the Hamiltonian in the truncated Fock space.
In Section 5 they study the obtained Hamiltonian in a mean-field approach. In Section 6 they perform an analytical description of the phase transition point using the mean-field Hamiltonian (59). In Section they account for finite-size effects via numerical fits.
The results presented here stand for a very fair amount of research.

I may have some remarks, some more serious than others, which may increase the readibility and the connection to current topical results.

1/I strongly believe that the term semiclassics is unappropriate. I can see that this is the main background of the authors but it seems to me they are performing here a standard (but not straightforward) mean-field analysis in order to describe the transition. Therefore I require to remove this term from the paper, because it is confusing, in order also to be more appealing to the condensed-matter community, or the cold-atom audience. In that spirit Ref.[21] is performing a mean-field description. This is here clearly the case: Section 5 follows a mean field approach.

2/The authors are very honnest in Section 1 when discussing the physical relevance of the model.
I am still not convinced that the proposed quench scenario will be indeed almost independent of the absolutely crucial truncation they do in the momentum space. This should be again stressed at the key step of deriving Eq.(8) where the truncation is implemented.

3/Related to 2/ the authors always use the term quantum phase transition. This usually refers to a change of properties of quantum ground state properties. Here they deal with a Hamiltonian, which is not bounded from below. Therefore I would also suggest to simply call this a phase transition, and maybe also relate it to the Excited State Quantum Phase transition community, which is rapidly growing, see e.g.:
Caprio, Cejnar, Iachello Annals of Physics (2008), instead of [34]
Brandes PRE (2013)

4/Can the described approach be able to predict some critical exponents, like the one shown in Eq.(72)?
To my understanding those are results from numerical fits. Do the authors conjecture a value 1/3 and 2/3? Can they justify it?

I have also some comments more specific to the text:
-- Abstract: "dynamical exponents characterizing the approach.." I understand that the authors do not mean the set of critical exponents, which are uniquely classifying the transition. Instead they are fitting numerical data and guessing some scaling laws in order to take into account the finite-size effects.
-- p.2, l.36: "a rigorous derivation": cite a math paper (Ref.[21] does not control the magnitude of error terms) or change the wording
-- p.3, Eq.(3): should the ket \psi be changed for k to be consistent with Eq.(2)?
-- Fig.3: Label the axis!
-- p.6: my understanding is that the approximation used here amounts to neglects the off-diagonal blocks of the decomposition in Eq.(21). Is it correct? Is there a control parameter for that? How do you justify it?
Can n0 be chosen in advance in a quench scenario to see which block plays a major role?
I guess this problem reappears in Eq.(24): do I understand correctly that the term H_coup is always neglected in the forthcoming analysis?
-- Fig. 4: I cannot see with the text why there are three red branches. Do they stand for several n0? If yes, please label them. Can the disagreement between black and green dots around alpha N\sim0.67 and E\sim4 be explained? Is the latter related to the phase transition?
-- Where does the second term in Eq.(50) come from? How is the front scalar coefficient derived/hinted?
-- I cannot clearly see the difference between Fig.4 and Fig.5. Could you please elaborate?
-- In Eq.(57) I do not understand the factor \cos(2\theta): Looking at Eq.(54) together with the definition of \theta does not lead to that as far as I can see
-- if L is taken to 0 effectively then the scale in energy does not make sense.

More technical details/misprints:
-- p.2,l.43: dispersion -> dispersion relation
-- There are several issues with the (confusing) notations:
Eq.(1): do not use alpha as as a sum index!,
Eq.(15): S should read \hat{S},
the choice of c_0 in Eq.(29) is a bit confusing (central charge?). Could you maybe use n with another font (\mathfrak{n})?,
the sum in expansion Eq.(25) is meaningless as afterwards, only H_1 on the one hand and H_0+H_2 are discussed,
Eq.(32-34): is \rho an index on the pseudo-spin as I suspect? Why not using \sigma?,
Ref [9] is incomplete.

Reviewer 2 Report

In this manuscript the authors show by analytical and numerical methods
that a pseudo relativistic extension of the Lieb-Liniger model of attractive
bosonic gases. The adiabatic separation of scales allows for an effective
model which in turn is studied by a mean field approach. With these tools
at hand, the authors predict a phase transition much in the same spirit
of the original model.

I think that in view of the recent interest in many body physics and
quantum chaos connections, this paper is a bold contribution to our
understanding of key phenomena in this realm. Thus, I recommend publication
in its present form.

Round 2

Reviewer 1 Report

First and more importantly I would like to thank the authors for their very accurate explanations and answers. I can now clearly better appreciate the (high) quality of their work.

I would like to make a very few further comments, which they might choose to consider:

- I am still not fully sure to understand what the x-axis of Fig.3 is. I strongly suspect they mean some (restricted list) of Fock states.
If it is actually n_0=(n_0+) + (n_0-) there should 120 points (like N). Is that correct?
It seems to me that this figure was an important step for the authors to build their intuition. That is the reason why it may be worth sharing how one can retrieve this intuition.

- the Ref.[21] only mentions briefly that the finite size scaling exponent (not sure dynamical exponent is the best label but I respect your decision) are computed in detail in a PhD manuscript, ref. [81] of the arxiv version. I think this is worth reminding where exactly to find this (impressive) result.

line 198: propper-> proper
